# Microbiome and Metabolite Analysis Insight into the Potential of Shrimp Head Hydrolysate to Alleviate Depression-like Behaviour in Growth-Period Mice Exposed to Chronic Stress

**DOI:** 10.3390/nu16121953

**Published:** 2024-06-19

**Authors:** Lianhua Hu, Weichang Ye, Qi Deng, Chen Wang, Jinjin Luo, Ling Huang, Zhijia Fang, Lijun Sun, Ravi Gooneratne

**Affiliations:** 1Guangdong Provincial Key Laboratory of Aquatic Product Processing and Safety, Guangdong Provincial Engineering Technology Research Center of Seafood, Guangdong Province Engineering Laboratory for Marine Biological Products, Key Laboratory of Advanced Processing of Aquatic Product of Guangdong Higher Education Institution, College of Food Science and Technology, Guangdong Ocean University, Zhanjiang 524088, China; hu-lian-hua@163.com (L.H.); yeweichang0_0@163.com (W.Y.); 18642326533@163.com (C.W.); 2112203018@stu.gdou.edu.cn (J.L.); 2112203048@stu.gdou.edu.cn (L.H.); fzj4437549@163.com (Z.F.); sunlijun@gdou.edu.cn (L.S.); 2Department of Wine, Food and Molecular Biosciences, Faculty of Agriculture and Life Sciences, Lincoln University, P.O. Box 85084, Lincoln 7647, New Zealand; ravi.gooneratne2@lincolnuni.ac.nz

**Keywords:** shrimp head hydrolysate, chronic stress during growth, gut microbiota, short-chain fatty acids, neurotransmitters

## Abstract

Chronic stress (CS) endangers the physical and mental health of adolescents. Therefore, alleviating and preventing such negative health impacts are a top priority. This study explores the effect of feeding shrimp head hydrolysate (SHH) on gut microbiota, short-chain fatty acids (SCFAs), and neurotransmitters in growing C57BL/6 mice subjected to chronic unpredictable mild stress. Mice in the model group and three SHH groups were exposed to CS for 44 days, distilled water and SHH doses of 0.18, 0.45, 0.90 g/kg·BW were given respectively by gavage daily for 30 days from the 15th day. The results showed that SHH can significantly reverse depression-like behaviour, amino acids degradation, α diversity and β diversity, proportion of Firmicutes and Bacteroidota, abundance of genera such as *Muribaculaceae*, *Bacteroides*, *Prevotellaceae_UCG-001*, *Parabacteroides* and *Alistipes*, concentration of five short-chain fatty acids (SCFAs), 5-HT and glutamate induced by CS. *Muribaculaceae* and butyric acid may be a controlled target. This study highlights the potential and broad application of SHH as an active ingredient in food to combat chronic stress damage.

## 1. Introduction

When the human body is stimulated by adverse external factors, it produces a stress response. Chronic stress (CS) refers to an individual experiencing long-term stress [1]. Prolonged stress response can disrupt the body’s internal homeostasis. CS may cause health issues that affect cardiovascular, endocrine, gastrointestinal, neurological and other systems resulting in anxiety, depression and cognitive impairment [1].

Adolescents encounter a variety of challenges such as demanding academic tasks, psychological pressure and sleep deprivation, all of which can lead to CS [2]. Due to the vulnerability of physical and mental functions during adolescence, major depressive disorder (MDD) is prevalent in the growing population with the prevalence of depression in adolescents growing faster than in adults [3,4,5,6,7,8]. In addition, CS during adolescence can lead to anxiety and depression in adulthood [4]. At present, there is a relative paucity of research in CS during adolescence. The current focuses on CS treatment are using drug intervention and it targets only the symptoms of depression. Current antidepressants not only appear to be more ineffective in treating depression in children and adolescents than in adults, but they may cause more side effects and increase the risk of suicide [9,10,11,12]. Therefore, it is necessary to explore more efficient and safe intervention methods.

In the case of CS, if selected foods could be used as a preventive measure to alleviate CS-related symptoms including depression, it would provide adolescents with safer health protection. Recent studies have revealed that some food-based active ingredients, such as polysaccharides, oligosaccharides, polyphenols and unsaturated fatty acids, have specific regulatory effects on some chronic stress-related health issues [13,14,15,16,17]. However, there is a lack of research on interventions for physiological and psychological function impairment in animals or humans experiencing CS during their growth period, which limits the development of products to improve CS. Therefore, it is necessary to explore the pathways and mechanisms by which foodborne intervention substances act on damage and adverse effects in growing model animals under CS.

Chronic stress can lead to a variety of negative effects on the body. There is increasing evidence that an imbalance in diversity and composition of gut microbiota and associated metabolites are decisive factors in CS-related dysfunction [18,19]. Some studies suggest that an imbalance in neurotransmitters such as 5-hydroxytryptamine (5-HT), dopamine and γ-aminobutyric acid (GABA) associated with gut microbiota changes can lead to depression induced by CS [20,21,22]. Short-chain fatty acids (SCFAs), especially mainly acetic acid, propionic acid and butyric acid, which are the main end products of bacterial fermentation in the gut, influence the intestinal barrier, neurotransmitters and inflammatory factors [23,24,25,26]. SCFAs are biomarkers of depression and key communication substances between the gut and brain, which can alleviate depression through the hypothalamic–pituitary–adrenal axis (HPA), tryptophan pathway and inflammatory response pathway [21,27,28,29]. Therefore, gut microbiota and its metabolites may be suggested as potential targets for intervening in adolescent CS. Our group found that SHH can effectively improve the gut microbiota disorder caused by capsaicin and antibiotics [30,31]. However, it is unclear whether SHH can alleviate depression-like behaviour by modulating intestinal microbiota structure and metabolic disturbances induced by CS in growing mice.

Therefore, to verify our hypothesis, the aim of the current study was to investigate the impact of SHH on gut microbiota, microbial metabolites and depression-like behaviour in a growing mice model under chronic stress. The effects of SHH on gut microbiota were evaluated based on diversity, species composition and differential microbiota; on gut microbiota metabolites from the perspectives of short-chain fatty acids and neurotransmitters. The correlations of amino acid metabolism, gut microbiota, metabolites and depression-like behaviour were conducted to understanding the approximate pathways of SHH to alleviate the disturbance induced by CS.

## 2. Materials and Methods

### 2.1. Shrimp Head Hydrolysate Preparation

The shrimp head was purchased from the Zhanjiang aquatic product market and was processed by removing the shell, followed by mashing and homogenization in distilled water in a 1:0.5 mass ratio. The pH was adjusted to 7.0, and neutral protease (5000 U/g) was added according to 2% shrimp head mass. After keeping it in a water bath at 50 °C for 3 h, the mixed solution was heated to 100 °C for 20 min to inactivate the enzyme. Next, the solution was cooled and centrifuged at 2455 rcf for 10 min. The top fat layer was removed, and the supernatant was freeze-dried for future use. The amino acid composition of SHH is shown in Table A1.

### 2.2. Animal Treatment

Three-week-old, healthy, specific pathogen-free (SPF) male C57BL/6J mice, with an average body mass of 10 ± 0.5 g, were purchased from Changsha Tianqin Biotechnology Co., Ltd. (Changsha, China) with the production licence number SCXK (Hunan) 2019-0014. The mice were reared in the SPF animal facility (licence number: SYKX [Guangdong] 2014-0053) at the School of Food Science and Technology, Guangdong Ocean University, at an average ambient temperature of 22 ± 1 °C, with 12 h light and dark cycle and free access to food and water.

Chronic unpredictable mild stress was induced using a variety of methods including loud noise for 30 min, cage tilt for 24 h, continuous light for 24 h, continuous darkness for 24 h, 12 °C water bath for 6 min, suspension for 30 min, fasting for 24 h and water deprivation for 24 h. One or two of these stress types were given daily. The same stress type was not repeated for two consecutive days. The specific plan is shown in Table A2.

The mice were randomly divided into six groups, a control (C) group, a model (M) group and low (0.18 g/kg·bw; XTL), medium (0.45 g/kg; XTM) and high (0.90 g/kg; XTH) groups given SHH, with six mice in each group. Except for Group C, the mice in all other groups were subjected to CS as described above for 14 days. In the next 30 days, Group C was administered distilled water by gavage without CS. Group M was subjected to CS and received distilled water by gavage for 30 days. Gavage frequency was once per day. The experimental plan is shown in a graphical format in Figure 1.

### 2.3. Samples Collection

On day 44, the faeces from each mouse were collected in a cryopreservation tube and stored in a −80 °C freezer.

### 2.4. Behavioural Tests

Behavioural tests commenced on day 44 [32].

#### 2.4.1. Sucrose Preference Test

This experiment lasted three days. On day 1, the drinking water was replaced with 1% sugar water. On day 2, the mice were given distilled water and 1% sugar water for 24 h and the position of the water bottle was changed every 12 h. On day 3, both water and 1% sucrose containing bottles were weighed and offered to mice for 24 h with the position of the bottle changed at 12 h. Next, the water and sucrose bottles were weighed. The preference for sucrose was calculated as (sucrose water consumption/total liquid consumption) × 100%.

#### 2.4.2. Elevated Plus Maze Test

The mice were placed in the center of an elevated maze (40 cm above the ground) which comprised two open arms (30 cm × 6 cm) and two closed arms (30 × 6 × 15 cm) with the head of the mice facing the open arm. The cumulative time the mice were in the open arm within 6 min was recorded.

#### 2.4.3. Open-Field Test

The mice were placed in an open-field box (40 × 40 × 40 cm) with the bottom of the box divided into 16 grids for 6 min. The peripheral 12 grids were defined as the peripheral area and the middle 4 grids as the central area. The cumulative time the mice were in the central area was recorded.

#### 2.4.4. Tail Suspension Test

The mouse tail was attached to a tail suspension scaffold using tape at 2 cm from the tail tip. This positioning resulted in the mouse head being suspended downward, preventing its limbs from contacting the surrounding objects for 6 min. The time each mouse was immobile within 4 min was recorded.

### 2.5. Gut Microbiota Measurements

The faecal samples were sent to Hangzhou Guhe Information Technology Co., Ltd. (Hangzhou, China) commissioned to perform high-throughput sequencing of the 16S rRNA gene using the Illumina NovaSeq6000 high-throughput sequencing platform (San Diego, CA, USA). The sequencing depth of samples is described in Table A3.

### 2.6. Short-Chain Fatty Acid Measurements

Detection and quantification of SCFAs in mouse faeces were performed using GC-MS. The chromatographic column was VF-1701 (30 m × 0.25 mm ID × 0.25 μm), with a split ratio of 5:1. The temperature of the ion source was 230 °C, and the ionisation mode was EI.

### 2.7. Neurotransmitter Measurements

Quantitative analyses of 5-HT, dopamine, glutamate and GABA in mouse faeces were performed by LC-MS using an Athena NH2 chromatographic column (2.1 mm × 150 mm) and a particle size of 5 μm. The mobile phase was a mixture of acetonitrile and a 0.1% aqueous solution of formic acid in a ratio of 2:8, and the flow rate was set at 0.3 mL/min. The mass spectrometer detector was ionised by electrospray ionisation and scanned in positive ion multiple reaction monitoring mode.

### 2.8. Statistical Analyses

#### 2.8.1. Microbiome Sequence

Raw sequencing reads with exact matches to the barcodes were assigned to respective samples and identified as valid sequences. Paired-end reads were assembled using Vsearch V2.22.1 (-fastq_mergepairs -fastq_minovlen 5). The Amplicon Sequence Variant (ASV) picking using Vsearch v2.22.1 was included in dereplication (-derep_fulllength), The performed quality control and denoising sequences with UNOISE2 algorithm (-cluster_unoise), Chimera removal (-uchime3_denovo) and mapping to ASVs were with a 100% similarity threshold (-usearch_global).

Due to the significant difference in the number of reads corresponding to each sample in bacterial 16S amplicon sequencing, to avoid bias in analysis caused by different sample data sizes, it was necessary to randomly flatten each sample. Generally, the minimum number of reads in the sequencing sample was selected as the base, which meant that the reads of all samples were uniformly flattened to this value. To minimise the differences in the sequencing depth across samples, normalization was performed on the ASV sequence counts of samples in the ASV Table, to ensure that the sum of ASV sequences in each sample was consistent. Normalised values were set to 1, representing relative abundances.

A representative sequence (rep-seqs) was selected from each ASV using default parameters. rep-seqs and ASV table files were imported into QIIME2 (QIIME2-2022.2). ASVs containing less than 0.001% of total sequences across all samples were discarded by QIIME2. Taxonomic identifications represented the respective sequences assigned to ASVs against the QIIME2 weighted taxonomic classifiers (silva-138-99-nb-weighted-classifier). The generated taxonomy was collapsed (Levels 1–7) using the “qiime taxa collapse” command. The “qiime taxa collapse” command was used to aggregate the ASV feature table to different classification levels. At each classification level, the community composition of each sample was calculated based on the kingdom, phylum, class, order, family, genus, and species. The abundance and classification of all ASVs in each sample were recorded. ASVs with a content lower than 0.001% of the total sequence in all samples were removed.

#### 2.8.2. Alpha and Beta Diversity

Alpha and beta diversity calculations were generated using the q2-diversity core-metrics phylogenetic tool from an ASV table.

Alpha-diversity metrics included Chao1 richness estimator, ACE metric (Abundance-based Coverage Estimator), Shannon diversity index and Simpson index, with statistically significant differences between groups identified using the Kruskal–Wallis test.

Beta diversity analysis was visualised via principal coordinate analysis (PCoA), principal component analysis (PCA) and non-metric multidimensional scaling (NMDS). NMDS was used for visualisation. It is also an analysis method based on sample distance matrix, which displays the specific distance distribution of samples through dimensionality reduction. NMDS analysis does not rely on the calculation of feature roots and eigenvectors. Instead, it ranks the samples by distance, making the sorting of samples in low dimensional space as close to each other as possible (rather than exact distance values). Each point in the diagram represents a sample, and points of the same colour come from the same group. Distance reflects sample similarity. NMDS evaluates inter-group differences through the stress function value of NMDS. When stress was <0.2, the results of NMDS analysis provided explanatory significance. When stress was <0.1, it could be considered a good sorting result. When stress was <0.05, it indicated that the analytical results denote excellent representativeness.

#### 2.8.3. Others

Statistical analyses were performed using SPSS 26.0. The results are expressed as the mean ± standard deviation (sd). ANOVA analysis was used to analyse the differences between groups. *p* < 0.05 or *p* < 0.01 served as the significant test levels. Pearson’s method was used to analyse the data correlation.

## 3. Results

### 3.1. Regulation of SHH on Behavioural Indexes

In the CS model, depression-like behaviour was evident. Key tests performed included the sucrose preference test, the open-field test, the elevated plus maze test and the tail suspension test (Figure 2). These tests were used to measure the degree of anhedonia, desire to explore and the degree of despair, all of which are symptoms of depression.

The behavioural indices of mice in the M group were significantly different from those in the C group (*p* < 0.05) and showed a lack of pleasure, decreased desire to explore and increased despair indicating CS induced depression-like behaviour in these mice. In addition to sucrose preference, SHH improved the depression-like behaviour of stressed mice, and the XTL group was the most affected.

Depression-like behaviour is closely related to gut microbiota and its metabolite changes [18,19]. In this study, SHH alleviated some depression-like behaviours induced by CS. It can also improve the gut microbiota proportions and alleviate metabolite imbalance, a disorder of chronically stressed mice.

### 3.2. Effect of SHH on Gut Microbiota

#### 3.2.1. Regulation of α-Diversity

Alpha diversity reflects the richness and evenness of microbial communities in an ecosystem (Figure 3). The Shannon index and the Simpson index were used to determine the microbial diversity in samples; the larger the value, the higher the community diversity. The Chao1 index and the Ace index were used to estimate the number of Operational Taxonomic Units (OTUs) in the samples. The larger the value, the higher the bacterial richness.

The Shannon and Simpson indices of α diversity in the M group were significantly increased (*p* < 0.05) compared to the C group, and both the low and medium doses of SHH induced significant effects on both these indices (*p* < 0.05). This showed that CS induced a gut microbiota disorder with an increase in diversity, while SHH alleviated the gut microbiota disorder by restoring diversity. Low dose had the best regulatory effect on the Shannon and Simpson indices (not different from the control group, *p* > 0.05). CS exhibited no significant effect on the richness of gut microbiota, and the regulation by SHH was minimal.

#### 3.2.2. Regulation of β Diversity

The values of β diversity, which is a measure of the similarity or dissimilarity of two bacterial communities, differed from one another. Non-metric Multidimensional Scaling (NMDS) analysis is a distance-based ordination technique. The NMDS chart (Figure 4) showed that the distribution of bacterial communities in the M group was far distant from the other groups, indicating that the species composition was quite different. Meanwhile, the C group and the three SHH groups had similar distribution distances, which meant the species composition was similar.

These results indicate that CS induced a significant change in the gut microbiota distribution in mice and that SHH was able to maintain a similar microbiota distribution to that in the control group.

#### 3.2.3. Regulation of Species Composition and Differential Microbiota

Phylum level

Firmicutes and Bacteroidota were the dominant phyla in the gut of the C group, accounting for more than 95% of the total percentage in each group, while the total percentage of the other groups was less than 5% (Figure 5).

The relative abundance of Firmicutes and Bacteroidota differed significantly between the M and the C groups. Specifically, the M group exhibited significantly higher Firmicutes and a lower Bacteroidota abundance (*p* < 0.01). When three SHH doses were given to CS mice, Firmicutes and Bacteroidota levels changed significantly (*p* < 0.01) with values similar to that of the C group. The Firmicute/Bacteriodota ratio in M groups was 2.2~4.2 times higher than in the other four groups, suggesting that SHH provided post-therapy protection (Figure 6).

Genus level

Among the top ten abundant genera, *Muribaculaceae*, *Bacteroides*, *Prevotellaceae_UCG-001*, *Parabacteroides* and *Alistipes* in the M group were the most significantly different from those in C and SHH groups (Figure 7 and Figure 8). The differential analysis (Figure 8) revealed that the M group consisted of 18 genera. Among these genera, 16 showed a significant increase while *Muribaculaceae* and *Prevotellaceae_UCG-001* showed significant decreases; then, SHH restored the abundance of these genera somewhat similar to C. The relative abundance of *Muribaculaceae* in group C was 59.9% with a decrease of 51.6% in the M group.

Based on the results of species composition and differential analysis between groups at the phylum and genus levels, it was evident that CS induced gut microbiota disorder in growing mice and SHH was able to reverse this disorder.

### 3.3. Regulation of SHH on Gut Microbiota Metabolites

#### 3.3.1. Metabolic Functional Gene Prediction

In PICRUST bioinformatics function prediction based on 16S rDNA sequence, the Gut–Brain Module (GBM) is used for predicting the functions of gut microbial-derived neurotransmitters. The Gut Metabolic Module (GMM) is a module that predicts metabolic function. Gut microbiota play a role via its metabolites. By prediction and analysis of the metabolic function of gut microbiota, it was evident that the metabolic pathways related to the metabolism of SCFAs such as propionic acid synthesis, butyric acid synthesis, isovaleric acid synthesis and neurotransmitters such as glutamic acid synthesis and GABA synthesis significantly increased (*p* < 0.05) in the M group (Figure 9). Moreover, the degradation of 19 amino acids in the M group significantly increased (*p* < 0.05) (Figure 10). Aspartate, glycine, arginine, lysine and alanine, which were more active in degradation metabolism, were abundant in SHH (Figure 10, Table A1). SHH induced a significant inhibitory effect in the metabolic changes induced by CS.

#### 3.3.2. Regulation of Short-Chain Fatty Acids

The results of SCFAs (Figure 11) showed that acetic acid, propionic acid, butyric acid, isovaleric acid and valeric acid concentrations in the M group increased significantly (*p* < 0.05). This increase was inhibited in the three SHH dose groups, thus maintaining it at around the level of the C group. The results are consistent with the prediction results for metabolic functional genes.

#### 3.3.3. Regulation of Neurotransmitters

The faecal 5-HT concentration in the M group increased (*p* < 0.05), while dopamine decreased (*p* < 0.05) (Figure 12). SHH was able to restore these neurotransmitters to the control group level (*p* < 0.05). The glutamate concentration was significantly increased (*p* < 0.05), while GABA decreased markedly (*p* < 0.05). The SHH was able to change neither of these levels (*p* < 0.05). The glutamate and GABA concentrations were not consistent with the prediction of metabolic gene function.

The actual SCFAs and neurotransmitter concentrations were basically consistent with the prediction that CS induces gut microbiota and its metabolite changes, and SHH improves the acetic acid, propionic acid, butyric acid, isovaleric acid, valeric acid, 5-HT and dopamine concentrations but not glutamate and GABA.

### 3.4. Correlation Analysis

#### 3.4.1. Correlation Analysis of Metabolites and Behavioural Indexes

In general, the metabolites of gut microbiota can impact the host behaviour. Correlation analysis of metabolites and behavioural indexes (Figure 13) suggested that butyric acid was most closely related to depression-like behaviour, next were valeric acid and GABA. Butyric acid was negatively correlated with the desire to explore and correlated with the degree of despair. Valproic acid was negatively correlated positively with exploratory desire and hedonia. GABA was positively correlated with exploratory desire and hedonia. However, there was no significant correlation between acetic acid, isobutyric acid, isovaleric acid, 5-HT, dopamine and behaviour.

#### 3.4.2. Correlation Analysis of Differential Flora and Metabolites

SCFAs and neurotransmitters are generally influenced by gut microbiota. The different bacterial genera are mainly related to SCFA, 5-HT and dopamine (Figure 14). Except for *Muribaculaceae* and *Prevotellaceae_UCG-001*, which are the opposite, most other bacterial genera are positively correlated with acetic acid, propionic acid, butyric acid, isovaleric acid and 5-HT. The correlation between dopamine and differential microbiota was almost opposite to that of 5-HT. Isobutyric acid, glutamic acid, and GABA were almost unrelated to differential bacterial communities.

#### 3.4.3. Correlation Analysis of SCFAs and Neurotransmitters

There may also be mutual influence between SCFAs and neurotransmitters. 5-HT is positively correlated with butyric acid and isovaleric acid, while dopamine is negatively correlated with acetic acid, propionic acid, butyric acid and isovaleric acid (Figure 15).

#### 3.4.4. Correlation Analysis of Amino Acid Metabolism and Differential Flora, SCFAs

GMM analysis suggest that SHH can significantly regulate 19 amino acid degradation to normal levels (Figure 10). It is necessary to analyse whether amino acid metabolism is related to significantly different bacterial genera and SCFAs. The correlation analysis results showed that almost all differential bacterial genera and SCFAs were positively correlated with arginine, lysine, leucine, alanine, valine, isoleucine, threonine, serine, proline, histidine, methionine, cysteine, tryptophan and glutamine degradation except *Muribaculaceae* and *Prevotellaceae_UCG-001* (Figure 16). *Muribaculaceae* was negatively correlated with mucin degradation.

## 4. Discussion

In this study, SHH was able to effectively improve the depression-like behaviour of mice influenced by CS, especially the exploration of desire and despair, indicating that SHH has the potential to reduce the harm caused by CS. However, SHH had no effect on improving anhedonia. The reason for this needs to be explored in future research.

Depression-like behaviour may be influenced by gut microbiota and its metabolites, but gut microbiota generally does not directly exert physiological effects. It is more likely that the effects are exerted via affecting some metabolic products such as SCFAs and neurotransmitters. Correlation analysis results indicate that SCFAs may be the key factor affecting the desire to explore and degree of despair, while 5-HT and dopamine do not directly affect behavioural indicators (Figure 13) due to the fact that the 5-HT and dopamine produced in the gut may not be absorbed and utilised. However, 5-HT and dopamine may have an impact on SCFAs as there is a positive correlation between 5-HT and butyric and isovaleric acid (Figure 15). Combining Section 3.1 (Figure 2) and Section 3.3.3 (Figure 12), SHH has no significant regulatory effect on sugar preference, glutamic acid and GABA. Therefore, the key to alleviating the desire to explore and degree of despair by SHH may be SCFAs, especially butyric acid.

The SCFAs levels can be directly affected by SCFAs producing bacteria. In this study, excessive SCFAs concentrations may be related to the significant increase in *Bacteroides*, *Parabacteroides*, *Alistipes*, *Lachnoclostridium*, *Colidextribacter*, *GCA-900066575*, *Oscillibacter*, *Dorea*, *Intestinimonas*, *Peptococcus*, which have the capacity to produce SCFAs [33,34,35,36,37,38,39,40,41,42,43]. According to the results in Section 3.2, the most significant change was observed in *Muribaculaceae*. The significant reduction in *Muribaculaceae* provided space for the growth and reproduction of other bacterial communities. The recovery of *Muribaculaceae* and SCFAs producing bacteria regulated by SHH may be an important pathway to improve depression-like behaviour.

Under the action of anaerobic bacteria, glycine, threonine, glutamate, lysine, ornithine and aspartate can be metabolised to acetate; propionate mainly produced from threonine, threonine, glutamate and lysine can be utilised for butyrate synthesis; valine, leucine and isoleucine, proline are used for the synthesis of isobutyric acid, isovaleric acid, and valeric acid, respectively [44,45]. It is reasonable to believe that when SCFAs are over-produced, the amino acids in the intestine are largely consumed by the SCFAs producing bacteria resulting in a lesser amount of available amino acids to other microbial growth and mucin synthesis. *Muribaculaceae* have been identified as main mucus degraders [46], which was negatively correlated with both mucin degradation and almost all amino acid degradation (Figure 16). It can be inferred that the decrease in *Muribaculaceae* is due to the reduction in mucin caused by amino acid consumption.

SHH is rich in various amino acids such as glutamate, aspartic acid, arginine, lysine, leucine, alanine, etc. (Table A1). It is possible that SHH can effectively supplement the consumed amino acids to restore the balance of amino acids and mucin degradation, then alleviate the disorder of gut microbiota, stabilising the SCFA levels, resulting in the improvement of depression-like behaviour in growing CS mice. It was apparent that peptides and amino acids in SHH could also be used by SCFA-producing bacteria, which are as important as polysaccharides. This finding challenged previous studies which reported that 90% of SCFAs are produced by intestinal undigested carbohydrate fermentation [47].

Currently, it is widely believed that high levels of SCFAs are beneficial to health while low levels are harmful. Some studies have reported that SCFA concentrations in adult CUMS mice/rats can significantly decline, particularly butyric acid [48,49,50]. On the contrary, our study showed significantly high levels of SCFAs except isobutyric acid in the M group accompanied by the gut microbiota disorder and depression-like behaviour, particularly butyric acid. This indicates that excessive concentrations of SCFAs are not necessarily more beneficial to health. Although the effect of SCFAs remains to be confirmed, this study suggests that excessive SCFAs, especially butyric acid, may be a key factor leading to depression-like behaviour in growing mice. Butyric acid may be a target for SHH to improve depression-like behaviour.

In addition, no significant changes in *Muribaculaceae* were found in the study of chronic unpredictable mild stress (CUMS) and depression models in adult mice/rats, indicating that *Muribaculaceae* may be one of the characteristic genera used to distinguish the gut microbiota disorder between adult mice with or without CS.

In addition, this study also found different characteristics between growing and adult mice/rat under chronic stress. In CS adult mice/rats, α diversity is reduced [51,52,53] and the abundance of Firmicutes is significantly decreased while the abundance of Bacteroidota is significantly increased [48,51]. However, our study showed the opposite effect. This difference could be explained by the fact that the gut microbiota of adolescents may be more susceptible to external factors than adults. The SHH was able to exert and had a significant inhibitory effect on the increased α diversity induced by CS. Our previous research found that SHH exhibited a significant restorative effect on the reduced α-diversity and the changed gut microbiota caused by capsaicin in mice [30]. Therefore, the SHH regulation of gut microbiota is not a single increase or decrease, but rather the ability to maintain a normal level and a balance of gut microbiota.

The proportion of Firmicutes and Bacteroides in microbiota is different in depression and healthy patients [54]. This result is supported by previous studies reporting that higher F/B ratio is positively correlated with the homeostasis of the gut microbiota [55]. In this study, the F/B value of the M group was 4.2 times higher than that of the C group, but the M group exhibited a disordered state, suggesting that a higher F/B value does not necessarily indicate better outcomes. the richness of the microbiota should be in a suitable range to maintain intestinal microbial homeostasis.

This study analysed the data only from a perspective of correlation. Gaps on whether and how SCFAs, especially butyric acid, affect depression-like behaviour induced by chronic stress need to be examined. Specific regulatory relationship between gut microbiota and its metabolites remains to be confirmed. Further investigation is required to explore the role of SHH in the regulation of CS, including a more in-depth examination of its specific components, the relationship and underlying mechanism of specific components on gut microbiota and metabolites.

## 5. Conclusions

SHH was able to reduce depression-like behaviour and regulate the intestinal homeostasis of mice exposed to CS during the growth phase. SHH was competent in restoring the α- and β-diversities, the ratio of Bacteroidota to Firmicutes and the abundance of key bacteria such as *Muribaculaceae*, *Bacteroides*, *Prevotellaceae_UCG-001*, *Parabacteroides* and *Alistipes*. SHH was also able to maintain the amino acid metabolism and microbiota metabolites including acetic, propionic, butyric, isovaleric and valeric acids, and 5-HT and dopamine at normal or near-normal levels. Excessive SCFAs may be due to the significantly increased SCFAs producing bacteria and vigorous amino acid metabolism. Butyric acid is highly relevant with behavioural indicators. Butyric acid and *Muribaculaceae* may be the target to alleviate depression-like behaviour in growth-period mice exposed to chronic stress. Amino acid metabolism may be the most important pathway to regulate butyric acid and *Muribaculaceae* by SHH. This study highlights the potential and broad application of SHH as an active supplement in food sources to combat CS damage.

## Figures and Tables

**Figure 1 nutrients-16-01953-f001:**
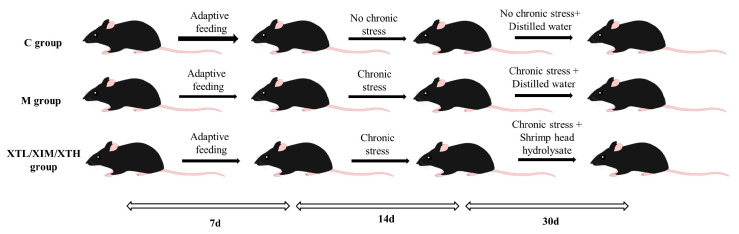
Experimental plan in a diagrammatical format. (C: control group; M: model group; XTL, XTM and XTH: low-, medium- and high-dose groups of shrimp head hydrolysate, respectively).

**Figure 2 nutrients-16-01953-f002:**
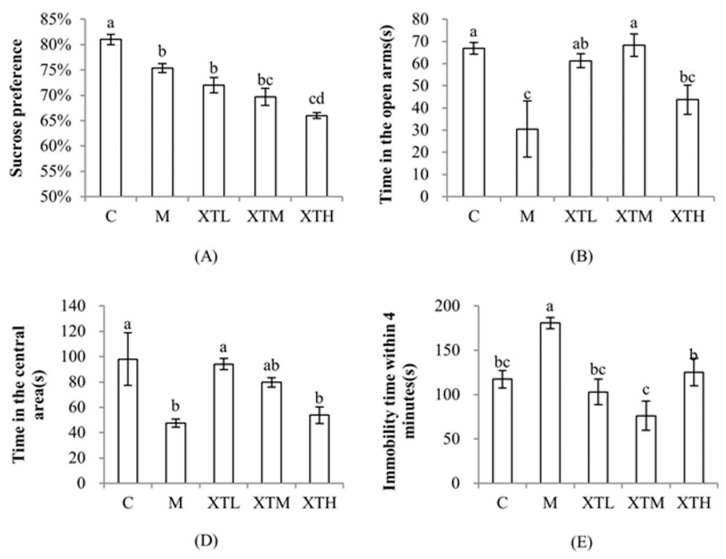
Comparison of behavioural indicators. (**A**): sucrose preference test; (**B**): open-field test; (**D**): elevated plus maze test; (**E**): tail suspension test. (C: control group; M: model group; XTL, XTM and XTH: low-, medium- and high-dose groups of shrimp head hydrolysate, respectively). Columns not sharing a common letter (a, b, c, d) are significantly different among the treatment groups (*p* < 0.05).

**Figure 3 nutrients-16-01953-f003:**
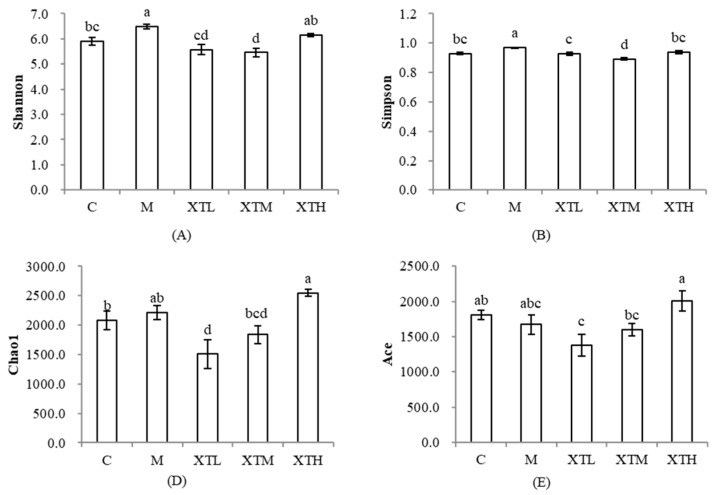
Comparison of α diversity differences. (**A**): Shannon index; (**B**): Simpson index; (**D**): Chao1 index; (**E**) Ace index. (C: control group; M: model group; XTL, XTM and XTH: low-, medium- and high-dose groups of shrimp head hydrolysate, respectively). Values not sharing a common letter (a, b, c, d) are significantly different between the groups (*p* < 0.05).

**Figure 4 nutrients-16-01953-f004:**
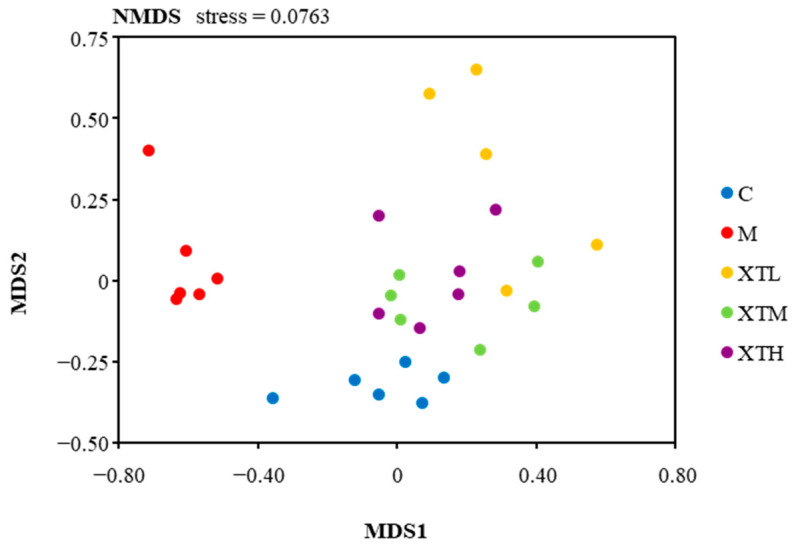
Non-metric Multidimensional Scaling (NMDS) analysis diagram. (C: control group; M: model group; XTL, XTM and XTH: low-, medium- and high-dose groups of shrimp head hydrolysate, respectively).

**Figure 5 nutrients-16-01953-f005:**
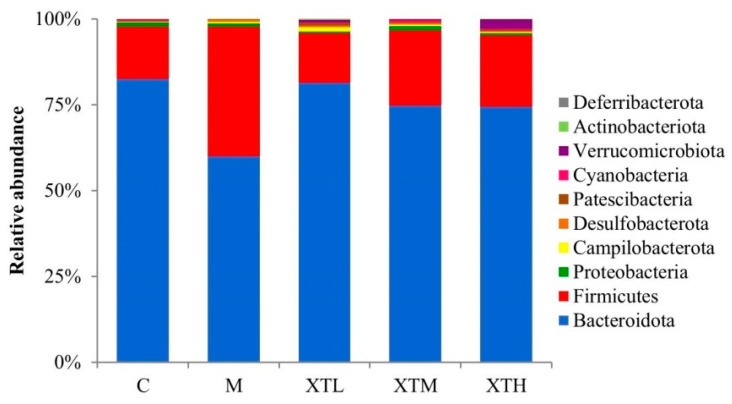
Species composition at the phylum level (C: control group; M: model group; XTL, XTM and XTH: low-, medium- and high-dose groups of shrimp head hydrolysate, respectively).

**Figure 6 nutrients-16-01953-f006:**
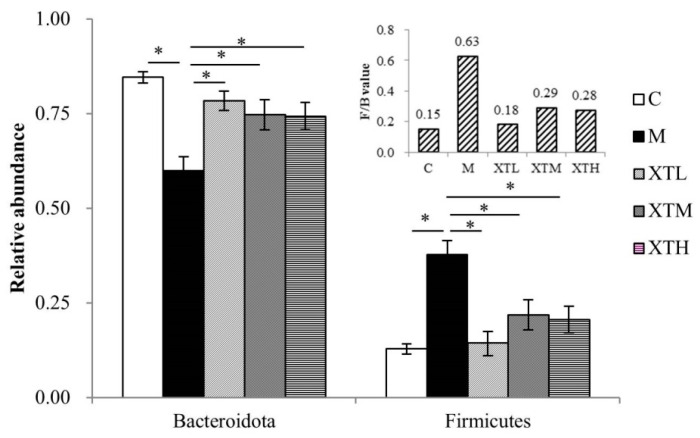
Comparison of differential gut microbiota at the phylum level (C: control group; M: model group; XTL, XTM and XTH: low-, medium- and high-dose groups of shrimp head hydrolysate, respectively) * Values not sharing a common letter are significantly different from those in the other groups (*p* < 0.01).

**Figure 7 nutrients-16-01953-f007:**
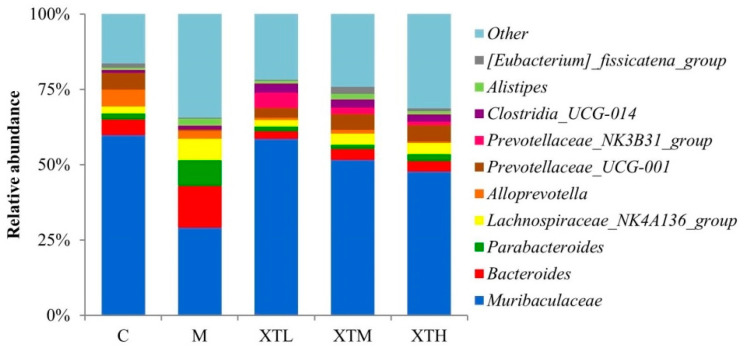
Top 10 species composition at the genus level (C: control group; M: model group; XTL, XTM and XTH: low-, medium- and high-dose groups of shrimp head hydrolysate, respectively).

**Figure 8 nutrients-16-01953-f008:**
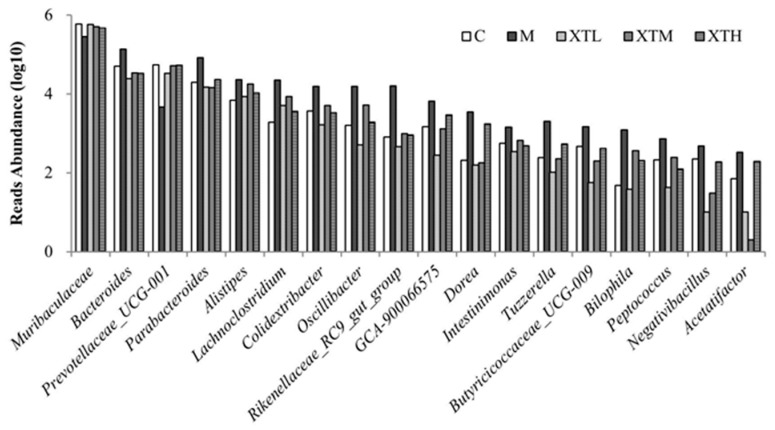
Comparison of differential gut microbiota at the genus level (*p* < 0.01 of all 18 genera) (C: control group; M: model group; XTL, XTM and XTH: low-, medium- and high-dose groups of shrimp head hydrolysate, respectively).

**Figure 9 nutrients-16-01953-f009:**
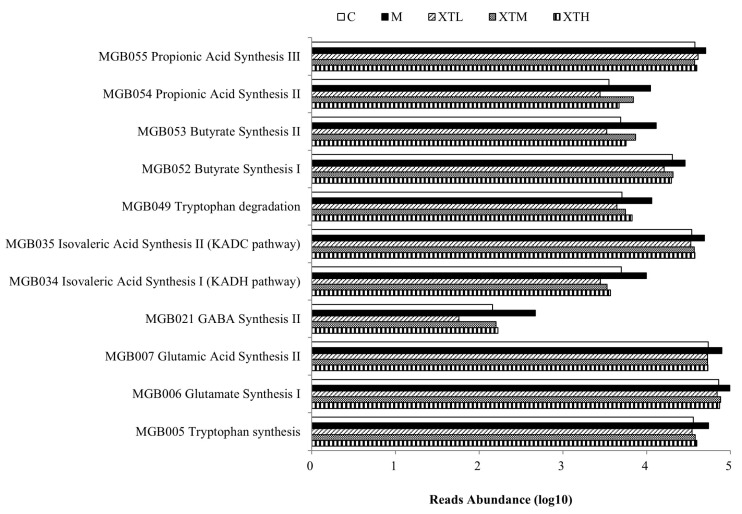
Prediction of differential GBM metabolic pathways related to short-chain fatty acids and neurotransmitters (*p* < 0.05) (C: control group; M: model group; XTL, XTM and XTH: low-, medium- and high-dose groups of shrimp head hydrolysate, respectively).

**Figure 10 nutrients-16-01953-f010:**
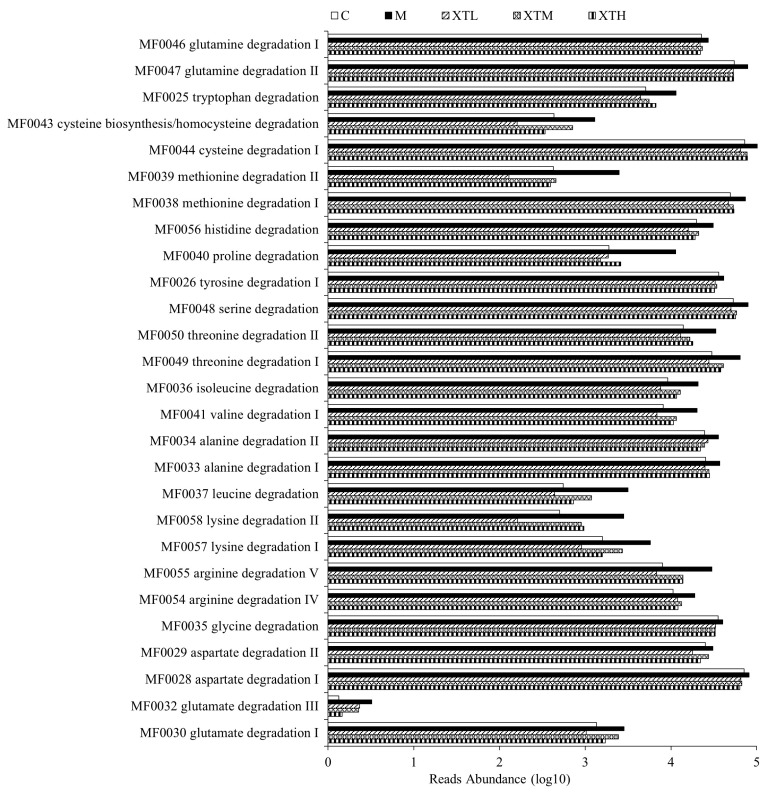
Prediction of differential GMM metabolic pathways related to amino acids (*p* < 0.05) (C: control group; M: model group; XTL, XTM and XTH: low-, medium- and high-dose groups of shrimp head hydrolysate, respectively).

**Figure 11 nutrients-16-01953-f011:**
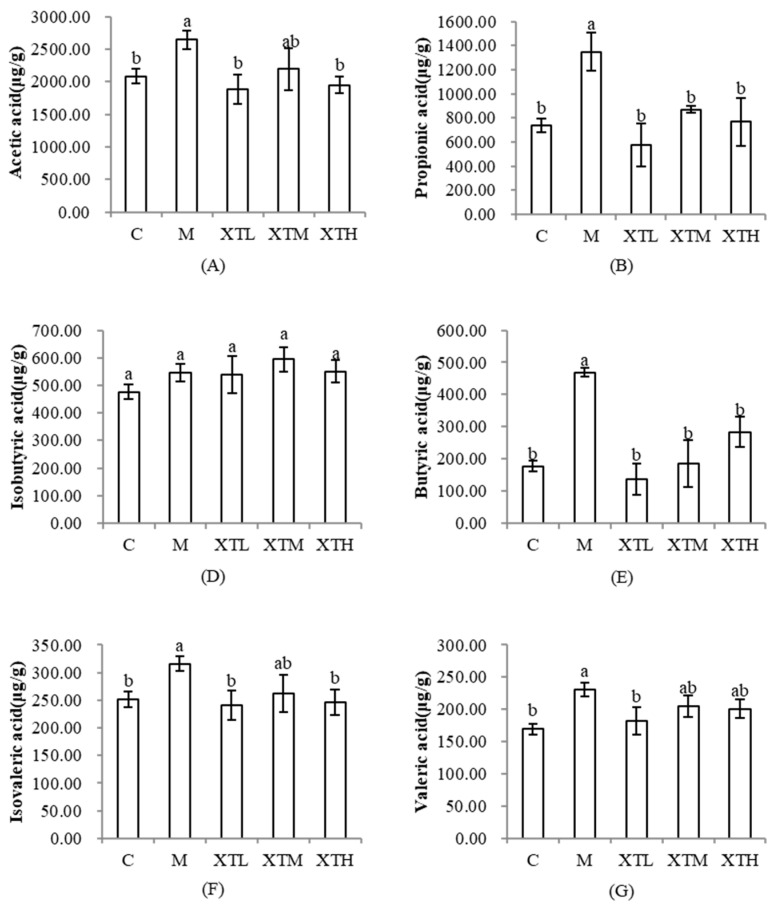
Comparison of short-chain fatty acid concentrations in each group (C: control group; M: model group; XTL, XTM and XTH: low-, medium- and high-dose groups of shrimp head hydrolysate, respectively). (**A**): acetic acid; (**B**): propionic acid; (**D**): isobutyric acid; (**E**): butyric acid; (**F**): isovaleric acid; (**G**): valeric acid. a, b values not sharing a common letter are significantly different between the groups (*p* < 0.05).

**Figure 12 nutrients-16-01953-f012:**
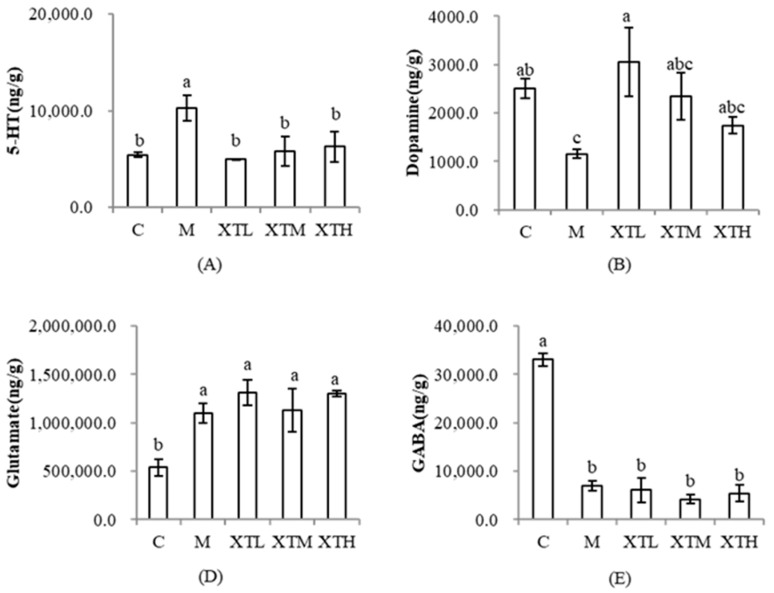
Comparison of neurotransmitter concentrations (C: control group; M: model group; XTL, XTM and XTH: low-, medium- and high-dose groups of shrimp head hydrolysate, respectively). (**A**): 5-HT; (**B**): dopamine; (**D**): glutamate; (**E**): GABA. a, b, c values sharing a common letter are not significantly different between the groups (*p* < 0.05).

**Figure 13 nutrients-16-01953-f013:**
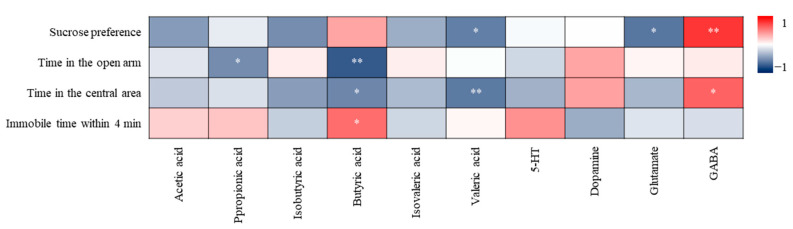
Correlation analysis of metabolites and behavioural indices. * Indicates a significant correlation at the 0.05 level; ** indicates a significant correlation at the 0.01 level.

**Figure 14 nutrients-16-01953-f014:**
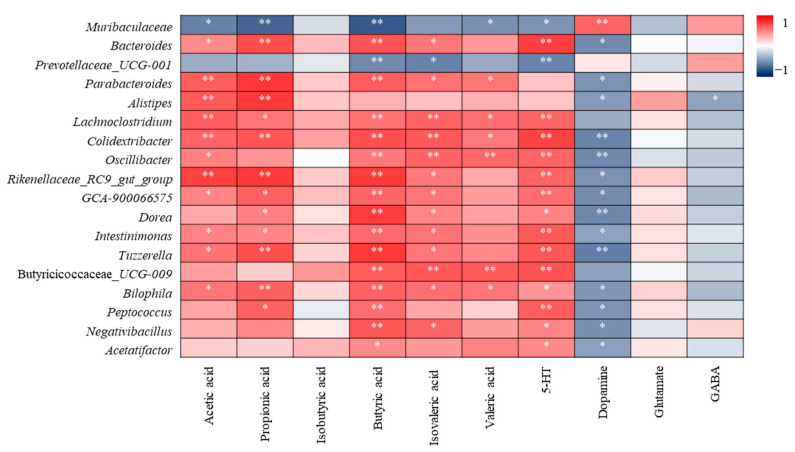
Correlation analysis of differential flora and metabolites. * Indicates a significant correlation at the 0.05 level; ** indicates a significant correlation at the 0.01 level.

**Figure 15 nutrients-16-01953-f015:**
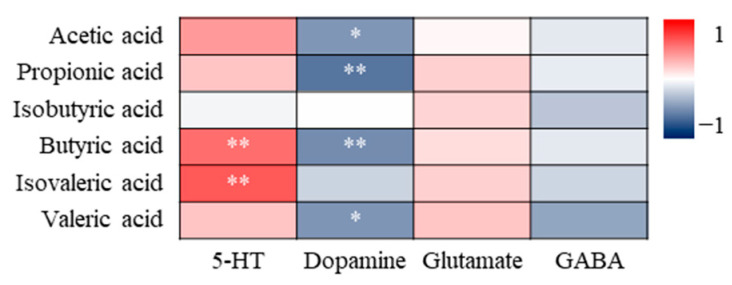
Correlation analysis of SCFAs and neurotransmitters. * Indicates a significant correlation at the 0.05 level; ** indicates a significant correlation at the 0.01 level.

**Figure 16 nutrients-16-01953-f016:**
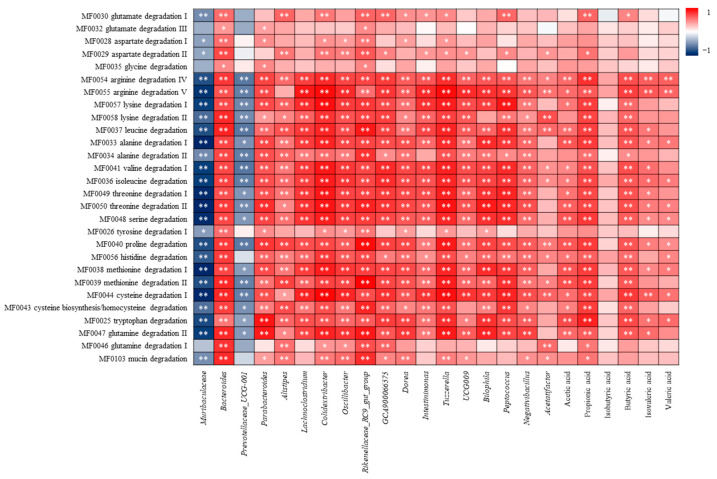
Correlation analysis of amino acid metabolism and differential flora, SCFAs. * Indicates a significant correlation at the 0.05 level; ** indicates a significant correlation at the 0.01 level.

## Data Availability

All data generated and/or analysed during the current study are available from the corresponding author upon reasonable request. Data are not available publicly due to privacy and intellectual property protection, as well as other legal and ethical reasons.

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
