# Peer review of "Microbiome and Metabolite Analysis Insight into the Potential of Shrimp Head Hydrolysate to Alleviate Depression-like Behaviour in Growth-Period Mice Exposed to Chronic Stress"

_nutrients, 2024, doi:10.3390/nu16121953_

Round 1

Reviewer 1 Report

Comments and Suggestions for Authors

In this study the author evaluate the effects of Chronic unpredictable mild stress on the depression and anxiety like behavior and the role of the gut microbiome, short-chain fatty acids (SCFAs), and neurotransmitters in causing the behavioral changes in mice. They also evaluate the effects of SHH on these parameters. The authors need to change several aspects of their manuscript in order to it to be ready for publication. 

Introduction

1- In the introduction the authors make several statements based on literature but fail to include the references. This needs to be addressed. One exemple: Line 52 Recent studies have revealed that some food-based active ingredients, such as polysaccharides, oligosaccharides, polyphenols and unsaturated fatty acids, have specific regulatory effects on some chronic stress-related health issues.

There a several instances in the introduction with the same problem.

2- The authors suggest the use of shrimp head hydrolysate to modulate the microbiota composition citing that in previous research it worked. However, they did not explain the rationale behind ii. What king of properties this substance has that can change the microbiota? They just gave it to the animals to see what happens?

Methods

3- The Chronic unpredictable mild stress  used in this study is vary hard to replicate. They do not say exactly which type of stress, and the number of stresses each animal was subjected during the experiments. On line 102 they say:  One or two of these stress types were given daily.

It would be very hard to replicate the results and even to analyze the results as One cannot say if all groups were subjected to the same stress types.

4- The authors need to elaborate longer in the statistical analysis. Specifically, they need to clarify how they analyzed the microbiome. How did they generate the alpha and beta indexes. Did they use PERMANOVA for the alpha index? 

Results

5- When showing Beta diversity the authors only show the NDMS chart but do not provide any statistical evidence that the communities are different. However they still say the communities are different. Statistical evidence must pre provided in order to claim such difference.

6- When showing the relative abundance the authors claim that relative abundance is different between groups. What does that mean? Considering this is a relative measurements the absolute number could be the same in the different groups. It is necessary to provide statistical evidence, such as Analysis of Composition of Microbiomes (ANCOM ),  to claim that  the number is different.

Discussion

7- The author fail to discuss the implication of the increased anedonia caused by the SHH.

8- The author use tests to measure anxiety but do not discuss the results based on this premiss.

9- The discussion needs to be rewritten according the results obtained by the new statistical analysis suggested

Comments on the Quality of English Language

The English needs to improve for clarity.

Reviewer 2 Report

Comments and Suggestions for Authors

Microbiome and metabolite analysis insight into the potential 2 

of shrimp head hydrolysate to alleviate depression-like behavior in growth-period mice exposed to chronic stress 

General comment

Authors investigated the effect of shrimp head hydrolysate (SHH) on CUMS induced depression symptoms, the microbiota, and metabolites. They found that SHH could ameliorate these phenotypes possibly by regulating fecal SCFA, BCAA, 5HT, and dopamine correlating with gut bacteria composition. Methods are based on established techniques and the interpretation of results seemed to have no logical defect. If the authors consider the minor points as described below, it will greatly improve the manuscript.

Specific comments

2.1. Shrimp head hydrolysate preparation L86, Table A1 L466

 Please add description about the content of lipids (especially carotenoids), and indigestible carbohydrate (such as glucosamine) in SHH.

2.2. Animal treatment L95

 Please describe mouse strain in more detail (C57BL/6J or N ?).

2.2. Animal treatment L107 

 Please describe gavage frequency (1 time / day ?) and energy ratio of SHH to daily intake of food.

2.5. Gut microbiota measurements L144

 Please describe the sequencing depth per sample.

Fig.8 L252

 Please add symbols for p<0.01 to the figure.

Fig. 14 L330, Discussion L419

 It is very interesting that the authors found clear correlation between fecal neurotransmitters and bacteria genera. If the authors perform correlational analysis among bacteria genera (e.g. Muribaculaceae vs. Bacteroides) and discuss their competitive or cooperative relationships mediated by nutrients (Trp, Asp, Tyr, Phe, and Glu), it may increase the value of manuscript.

References 20 L517, 21 L519

It is not reader-friendly to refer Chinese journals in the manuscript submitted to international journal.

Round 2

Reviewer 1 Report

Comments and Suggestions for Authors

The paper improved enough to be accepted